# Sex Difference in Control of Low-Density Lipoprotein Cholesterol in Older Patients after Acute Coronary Syndrome

**DOI:** 10.3390/geriatrics7040071

**Published:** 2022-06-24

**Authors:** Tan Van Nguyen, Dieu Thi Thanh Tran, Trinh Thi Kim Ngo, Tu Ngoc Nguyen

**Affiliations:** 1Department of Geriatrics & Gerontology, University of Medicine and Pharmacy at Ho Chi Minh City, Ho Chi Minh City 700000, Vietnam; nguyenvtan10@ump.edu.vn; 2Department of Interventional Cardiology, Thong Nhat Hospital, Ho Chi Minh City 700000, Vietnam; 3Faculty of Pharmacy, University of Medicine and Pharmacy at Ho Chi Minh City, Ho Chi Minh City 700000, Vietnam; thanhdieubmt@gmail.com; 4Faculty of Medicine, Nguyen Tat Thanh University, Ho Chi Minh City 700000, Vietnam; drkimtrinh115@gmail.com; 5Unit of Geriatric Medicine and Stroke Medicine, Ho Chi Minh City Orthopaedics and Rehabilitation Hospital, Ho Chi Minh City 700000, Vietnam; 6Westmead Applied Research Centre, Sydney Medical School, Faculty of Medicine and Health, The University of Sydney, Sydney, NSW 2145, Australia

**Keywords:** acute coronary syndrome, secondary prevention, lipid management, low-density lipoprotein cholesterol, elderly, frailty

## Abstract

Objective. The main aim of this study is to examine the achievement of low-density lipoprotein cholesterol (LDL-C) targets in older patients after acute coronary syndrome (ACS), and the secondary aim is to examine sex difference in LDL-C target achievement. Methods. Patients aged ≥60 years with ACS admitted to a tertiary hospital in Vietnam were recruited from December 2019 to August 2020. LDL-C target achievement was defined as having an LDL-C goal of <1.8 mmol/L. Multivariate logistic regression was applied to investigate the predictive factors for LDL-C target achievement. Results. A total of 232 participants were included in this study (mean age 75.5 years, 40.1% were women). Participants had an average of three chronic conditions other than coronary heart disease. All participants were prescribed statin monotherapy at discharge (59.5% on high-intensity statins). After 3 months, 218 (94.0%) of the participants were on statin monotherapy, 14 (6.0%) were on statin combined with ezetimibe. The proportion of participants that achieved LDL-C target after 3 months was 56.5% (40.9% in women and 66.9% in men, *p* < 0.001). On univariate logistic regression, women were less likely to achieve their LDL-C target compared to men (unadjusted OR 0.34, 95% CI 0.20–0.59). This association was still significant in the adjusted model (adjusted OR 0.43, 95% CI 0.24–0.78). Other factors that were significantly associated with LDL-C target achievement included age, smoking, sedentary lifestyle, LDL-C level on admission, history of using statin before admission, and high-intensity statin prescribed at discharge. Conclusions. Our study found that nearly a half of older patients with ACS did not achieve their LDL-C target after 3 months, and suboptimal control of LDL-C was more common in women.

## 1. Introduction

Coronary heart disease (CHD) is the leading cause of mortality globally [1,2]. Dyslipidemia is one of the most important factors in the pathophysiology of acute coronary syndrome, and the retention of low-density lipoprotein cholesterol (LDL-C) inside the arterial wall is the key initiating event in atherogenesis [3]. Evidence from clinical trials indicated that lower LDL-C values are associated with lower risk of future cardiovascular events and that statin use in patients after acute coronary syndrome (ACS) can improve clinical outcomes [4,5].

The prevalence of coronary heart disease increases with age, and older patients account for a large proportion of patients presenting with acute coronary syndrome [6]. Although cardiovascular disease is the leading cause of mortality in both men and women, women are less likely to be on secondary prevention treatments as they get older, and studies have shown higher risk of mortality and side effects from cardiovascular therapies in women compared to men [7]. The menopausal transition may contribute to changes in cardiovascular risk in women, including changes in body fat distribution and vascular inflammation [8]. In Vietnam, cardiovascular disease is the leading cause of mortality, and coronary heart disease is common in older people [9,10,11,12]. Understanding the status of achieving treatment goals is important to inform clinical practice, especially in older and frail patients. However, there is limited evidence on the achievement of LDL-C targets in older patients with coronary heart disease in Vietnam. Therefore, in this study on older patients with acute coronary syndrome, we aimed to investigate the proportion of patients attaining LDL-C targets at three months after discharge. The secondary aim was to examine sex differences in LDL-C target achievement in this population.

## 2. Methods

### 2.1. Study Design and Population

This was a prospective observational study at the Department of Interventional Cardiology, Thong Nhat Hospital in Ho Chi Minh City, Vietnam, from December 2019 to August 2020. Inclusion criteria included (1) age ≥60 years; (2) being diagnosed with ACS on admission, including ST-segment elevation myocardial infarction (STEMI), non-ST elevation myocardial infarction (NSTEMI), and unstable angina; and (3) being prescribed with statins at discharge. Patients were excluded from this study if (1) they did not provide consent; (2) they died during hospitalization.

The study was approved by the Ethics Committee of Thong Nhat Hospital (33/2019/BVTN-HDYD). Written informed consent was obtained from all participants.

### 2.2. Sample Size Calculation

The sample size was determined using a single population proportion formula: *n* = (1.96/d)^2^ × p × (1 − p), with *n* = the required sample size, p = proportion of patients achieving LDL-C target, and d = precision (assumed as 0.07). Taking references from reports from China, Singapore, and Thailand [13,14,15], we estimated that the proportion of patients achieving LDL-C target would be around 35%. Therefore, the sample size for this study was calculated to be at least 216 participants (allowing for 20% drop-out in three months of follow up).

### 2.3. Data Collection

Data were collected from patient interviews and from medical records. Information obtained included demographic characteristics, height, weight, medical history, history of using statin before this admission, comorbidities, frailty, admission diagnosis, laboratory results during hospitalization including lipid profiles, and medications prescribed upon discharge.

All participants were invited to attend a visit at the study clinics at the end of the third month after discharge. During this visit, participants were interviewed to obtain information about current medication use. Fasting serum lipid profile tests were conducted at this visit.

### 2.4. Variable Definition

#### 2.4.1. Acute Coronary Syndrome (ACS)

In this study, ACS referred to these three conditions [16]:
(1)ST-segment elevation myocardial infarction (STEMI): patients with acute chest pain and persistent (>20 min) ST-segment elevation on ECGs, with troponin changes.(2)Non-ST elevation myocardial infarction (NSTEMI): patients with acute chest discomfort but no persistent ST-segment elevation on ECGs, with troponin changes.(3)Unstable angina: unstable angina is defined as myocardial ischemia at rest or on minimal exertion in the absence of acute cardiomyocyte injury/necrosis (no troponin changes), without persistent ST-segment elevation.

#### 2.4.2. Outcome Variable

The outcome of interest was the proportion of patients achieving therapeutic LDL-C targets while on statin treatment after 3-month follow up, defined as an LDL-C goal of <1.8 mmol/L (<70 mg/dL) at the third month after discharge, as recommended by the 2015 guidelines for the diagnosis and management of dyslipidemia from the Vietnam Heart Association [17].

#### 2.4.3. Predictive Variables

##### Demographics and Lifestyle Factors

Age and sex were used as recorded in medical records. Body mass index (BMI, kg/m^2^) was calculated based on measured weight (kg) and height (m) and was classified into 4 groups: underweight (BMI < 18.5 kg/m^2^), normal (BMI 18.5–22.9 kg/m^2^), overweight (BMI 23–24.9 kg/m^2^), and obese (BMI ≥ 25.0 kg/m^2^). Smoking status was categorized based on self-report as non-smoking or smoking (including current smokers or ex-smokers who stopped smoking less than 1 year ago). Sedentary lifestyle was defined if participants did not do any regular physical exercise or walking for at least 20 min for at least 3 days a week.

##### Comorbidities

Comorbidities were obtained from medical records and were categorized into a list of 18 conditions: diabetes, obesity, hypertension, heart failure, stroke/transient ischemic attack, atrial fibrillation, peripheral artery disease, chronic kidney disease, chronic obstructive pulmonary disease, stomach problem, gout, osteoarthritis, cancer, anxiety, anemia, dementia, thyroid problem, urinary problem.

##### Frailty

Participant’s frailty status was defined according to the Clinical Frailty Scale (CFS) [18,19]. The CFS score ranges from 1 to 9, with a score of at least 5 indicating a frailty status [18,20].

##### Statin Prescription at Discharge

The type and dose of statins prescribed at discharge were documented. High-intensity statins were defined as rosuvastatin 20–40 mg or atorvastatin 40–80 mg [4].

### 2.5. Statistical Analysis

Analysis of the data was performed using SPSS for Windows 27.0 (IBM Corp., Armonk, NY, USA). Continuous variables are presented as means ± standard deviation, and categorical variables as frequencies and percentages. Comparisons between groups (achieving/non-achieving of LDL-C targets, men/women) were conducted using the chi-square test or Fisher’s exact test for categorical variables and Student’s t-test or Mann–Whitney test for continuous variables.

To examine the impact of sex on LDL-C target achievement after 3 months, multivariable logistic regression analysis was applied with sex as the independent variable of interest, adjusting for the factors that can affect LDL-C target achievement based on clinical rationale (such as age, smoking, sedentary lifestyle, overweight and obesity, comorbidities, frailty, history of using statin, statin strength, LDL level at baseline, and whether the participants received percutaneous coronary intervention or not).

All variables were examined for interaction and multicollinearity. Results are presented as odds ratios and 95% confidence intervals.

## 3. Results

### 3.1. Study Sample Characteristics

Table 1 presents the characteristics of participants stratified by achievement of LDL-C target. There were 264 participants who met the selection criteria, 32 participants were lost during the 3 months of follow up. Therefore, 232 participants were included in the analysis. They had a mean age of 75.5 years; 40.1% were women, and 64.2% were frail. Overall, participants had an average of three chronic conditions besides coronary heart disease, with no significant difference in the total number of comorbidities as well as in the prevalence of each disease between the two groups. The most common cardiometabolic comorbidities were hypertension (94.8%), heart failure (51.7%), and diabetes (45.7%). The most common non-cardiometabolic comorbidities were stomach disorder (61.9%) and chronic kidney disease (22.4%). The prevalence rates of overweight/obesity and of having sedentary lifestyle were high (Table 1).

### 3.2. Statin Utilization

Overall, 52.2% of the participants had a history of using statins before this admission, with no significant difference between those with and without LDL-C target achievement (46.6% and 59.4%, *p* = 0.052, respectively).

Details of statin prescription at discharge are presented in Table 2. Upon discharge, 59.5% of the participants were prescribed high-intensity statins, higher in participants achieving LDL-C target compared to participants who did not achieve LDL-C target (66.4% versus 50.5%, *p* = 0.014, respectively).

After 3 months, 218 (94.0%) of the participants were still on statin monotherapy, and 14 (6.0%) were on statin combined with ezetimibe (Table 3).

### 3.3. LDL-C Target Attainment

Overall, 56.5% (131/232) of the participants achieved the LDL-C target after 3 months. The median LDL-C level at admission for the entire cohort was 2.45 mmol/L (range 0.70–6.50), 2.9 mmol/L (range 0.80–6.24) in participants who did not achieve the LDL-C target versus 2.2 mmol/L (range 0.70–6.50) in participants achieving the LDL-C target, *p* < 0.001. After 3 months, the median LDL-C was 1.71 mmol/L (range 0.25–6.85) overall, 2.40 mmol/L (range 1.80–6.85) in the group that did not attain the LDL-C target, and 1.37 mmol/L (range 0.25–1.77) in the group that attained the LDL-C target, *p* < 0.001.

### 3.4. Sex Difference in Achieving LDL-C Target

Women had higher baseline LDL-C levels compared to men (median 2.60 mmol/L in women versus 2.33 mmol/L in men, respectively, *p* = 0.084). After 3 months, women still had higher LDL-C levels (median 2.00 mmol/L versus 1.60 mmol/L in men, *p* < 0.001) (Figure 1).

The proportion of women achieving their LDL-C targets (<1.8 mmol/L) after 3 months was lower compared to that of men (40.9% versus 66.9%, *p* < 0.001, respectively).

On univariate logistic regression, women were less likely to achieve their LDL-C targets compared to men (unadjusted OR 0.34, 95% CI 0.20–0.59). This association was still significant in the adjusted model, with an adjusted OR of 0.43 (95% CI 0.24–0.78). Other factors that had a significant impact on LDL-C target achievement included age, smoking, sedentary lifestyle, LDL-C level on admission, history of using statin before admission, and high-intensity statin prescribed at discharge (Table 4).

## 4. Discussion

In this study in 232 older patients with ACS, there was a suboptimal control of LDL-C, with only 56.5% of the participants achieved the LDL-C target after 3 months. 

Our findings are in line with reports from previous studies, in which less than 45% of patients with high cardiovascular risk could reach the recommended LDL-C targets [15,21,22,23,24,25]. According to the CEPHEUS study conducted in 35,121 participants (mean age 60.4 years) from 29 countries across Asia, Europe, the Middle East, and Africa, only 49.4% of the participants obtained their recommended LDL-C goal [22]. This proportion was particularly low in patients with very high cardiovascular risk (23%) compared to 55% in those with high CV risk, and 90% in those with low CV risk [22]. A study in 712 older patients (mean age 61.4 years) in Greece with prior coronary heart disease also reported that the LDL-C target was achieved in only 10% of the participants [23]. Another study in 267 patients with ACS (mean age 69) in Singapore showed that the LDL-C goal attainment rate was 36.7% at 4 months and declined progressively throughout follow up (23.4% at 12 months) [15].

We found that women were less likely to achieve their LDL-C targets compared to men. Sex differences in the effectiveness of statins have been reported in many studies. In a study in 14,710 statin users and 23,833 non-users discharged from hospital after an acute myocardial infarction in Canada, the effect of statin was reduced in women compared to that in men: for all-cause mortality within 1 year of follow up, the adjusted hazard ratios associated with statin use in women were 0.61 (95% CI 0.54–0.69) compared to 0.54 (0.48–0.60) in men, and for cardiac-related mortality within 1 year of follow-up, the adjusted hazard ratios associated with statin use in women were 0.70 (0.60–0.81) compared to 0.59 (0.51–0.69) in men [26]. In a study in 71 men and 166 women treated with different statins in Italy, there was a significantly attenuated reduction of total cholesterol and LDL-C in women compared to that in men after 1 year: −22.7% ± 11.8% in women versus −28.5% ± 11.8% in men vs. (*p* < 0.001, adjusted for dose and statin power) [27]. In another study in 415,294 patients (45.3% women) from 236 diabetes centers in Italy, women were less likely to reach LDL-C targets than men (45.6% versus 52.8% for women and men, respectively). Pooled analysis from 32,782 patients from 28 countries in the CEPHEUS study also showed that female gender was associated with a reduced likelihood of attaining LDL-C goals (adjusted OR 0.71, 95% CI 0.65–0.77) [22].

In addition to sex, we also observed that older age and high-intensity statin prescription at discharge were associated with an increased likelihood of achieving the target, while smoking, sedentary lifestyle, LDL-C level on admission, and history of using statin before admission were associated with a reduced likelihood of achieving the target. As smoking and sedentary lifestyle are modifiable factors, this result suggests that we need to pay more attention to risk factor management in older patients after ACS, particularly for smoking cessation and exercise training programs. There was a high prevalence of physical inactivity in Vietnam [28]. The smoking rates in Vietnam are also very high. According to the Vietnam Global Adult Tobacco Survey in 2015, 45.3% of Vietnamese men were current smokers [29]. In the era of percutaneous coronary intervention and optimal cardiovascular pharmacotherapy, smoking still has a significantly negative impact on survival after ASC. A study in 9375 patients with ACS from the Melbourne Interventional Group registry showed that persistent smoking was an independent predictor of increased mortality after ACS (HR 1.78, 95 CI 1.36–2.32, *p* < 0.001) [30]. A recent systematic review and meta-analysis in 11,228 patients from 39 studies revealed that the majority of smokers with an ACS continue to smoke after discharge and that attending cardiac rehabilitation increased the likelihood of smoking cessation of quitting (OR 1.90, 95% CI 1.44–2.51) [31]. Interestingly, participants with a history of using statin before admission were less likely to obtain LDL-C target achievement. More studies are needed to understand statin intolerance, statin resistance, and medication adherence in this group of patients.

Our study has several strengths. To our best knowledge, this is the first study investigating the achievement of LDL-C targets in older patients with ACS in Vietnam. This study was conducted at a large teaching hospital in Ho Chi Minh City, Vietnam, and contained high-quality, detailed clinical information. However, this study also has some limitations. Data were not available on medication adherence to statins. There was also limited information on other relevant factors related to LDL-C target achievement such as economic status and access to statins, the use of other medicines that can have an influence on lipid levels, and familial hypercholesterolemia. This study was conducted at only one hospital in Vietnam. Therefore, the studied population may not be representative of all older patients in Vietnam, and the results should be interpreted cautiously.

In conclusion, our study found that LDL-C control was suboptimal in nearly half of older patients with ACS and that women were less likely to achieve LDL-C target after 3 months compared to men. Greater efforts are needed to improve the secondary prevention of cardiovascular disease in women, and patients should also be assessed for smoking cessation and exercise training programs.

## Figures and Tables

**Figure 1 geriatrics-07-00071-f001:**
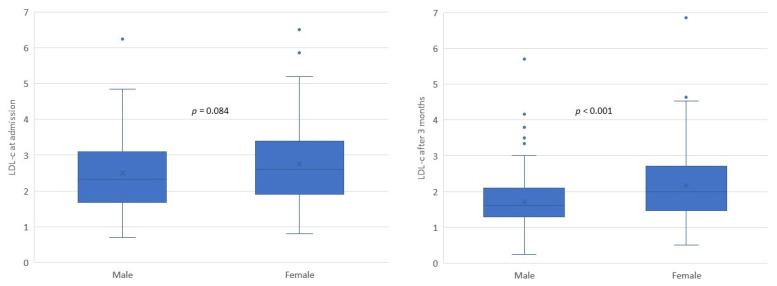
Low-density lipoprotein cholesterol (LDL-C) levels at admission and after 3 months in male and female participants.

**Table 1 geriatrics-07-00071-t001:** Participant characteristics.

Variables	All Participants (*n* = 232)
Age	75.4 ± 9.4
Sex:	
Female	93 (40.1)
Male	139 (59.9)
BMI:	
Underweight	20 (8.6)
Normal	111 (47.8)
Overweight	48 (20.7)
Obese	53 (22.8)
Sedentary lifestyle	101 (43.5)
Smoking	35 (15.1)
History of using statin before admission	121 (52.2)
History of myocardial infarction	40 (17.2)
History of percutaneous coronary intervention	65 (28.0)
Treated with percutaneous coronary intervention during admission	128 (55.2)
ACS types:	
STEMI	34 (14.7)
NSTEMI	122 (52.6)
Unstable angina	76 (32.8)
Frailty (CFS score ≥ 5)	149 (64.2)
Total number of comorbidities	3.16 ± 1.22
Cardiovascular comorbidities:	
Hypertension	220 (94.8)
Heart failure	120 (51.7)
Diabetes	106 (45.7)
Atrial fibrillation	18 (7.8)
Peripheral artery disease	17 (7.3)
Stroke/transient ischemic attack	13 (5.6)
Non-cardiovascular comorbidities:	
Stomach problem	143 (61.9)
Chronic kidney disease	52 (22.4)
Urinary problem	10 (4.3)
Cancer	9 (3.9)
Gout	8 (3.4)
Chronic obstructive pulmonary disease	7 (3.0)
Osteoarthritis	7 (3.0)
Anemia	6 (2.6)
Anxiety	4 (1.7)
Dementia	2 (0.9)
Thyroid problem	1 (0.4)

Continuous data are presented as mean ± standard deviation. Categorical data are shown as *n* (%). LDL-C, low-density lipoprotein cholesterol; BMI, body mass index; PCI, percutaneous coronary intervention; ACS, acute coronary syndrome; NSTEMI, non-ST segment elevation myocardial infarction; STEMI, ST-segment elevation myocardial infarction; CFS, Clinical Frailty Scale.

**Table 2 geriatrics-07-00071-t002:** Prescription of statin at discharge.

Variables	All Participants (*n* = 232)	Did Not Achieve LDL-C Target after 3 Months (*n* = 101)	Achieved LDL-C Target after 3 Months (*n* = 131)	*p*
Statin types:				
Atorvastatin 10 mg	8 (3.4)	4 (4.0)	4 (3.1)	0.101
Atorvastatin 20 mg	76 (32.8)	40 (39.6)	36 (27.5)	
Rosuvastatin 10 mg	10 (4.3)	6 (5.9)	4 (3.1)	
Rosuvastatin 20 mg	138 (59.5)	51 (50.5)	87 (66.4)	
Statin strength:				
High-intensity statins	138 (59.5)	51 (50.5)	87 (66.4)	0.014
Non-high-intensity statins	94 (40.5)	50 (49.5)	44 (33.6)	

Continuous data are presented as mean ± standard deviation. Categorical data are shown as *n* (%). LDL-C, low-density lipoprotein cholesterol.

**Table 3 geriatrics-07-00071-t003:** Statin use after 3-month follow up.

Variables	All Participants (*n* = 232)	Not Achieving LDL-C Target after 3 Months (*n* = 101)	Achieving LDL-C Target after 3 Months (*n* = 131)	*p*
Statin monotherapy	218 (94.0)	95 (94.1)	123 (93.9)	0.958
Statin plus ezetimibe	14 (6.0)	6 (5.9)	8 (6.1)	

Continuous data are presented as mean ± standard deviation. Categorical data are shown as *n* (%). LDL-C, low-density lipoprotein cholesterol.

**Table 4 geriatrics-07-00071-t004:** Univariate and multivariate logistic regression of sex and other factors on low-density lipoprotein cholesterol target achievement.

Factors	Univariate Analysis		Multivariate Analysis	
	Unadjusted Odds Ratios for LDL-C Target Achievement (95% CI)	*p*	Adjusted Odds Ratios for LDL-C Target Achievement (95% CI)	*p*
Female (vs. male)	0.34 (0.20–0.59)	<0.001	0.25 (0.13–0.51)	<0.001
Age	0.98 (0.95–1.01)	0.148	1.07 (1.02–1.13)	0.011
Smoking	0.25 (0.11–0.55)	<0.001	0.10 (0.04–0.29)	<0.001
Having sedentary lifestyle	0.27 (0.15–0.46)	<0.001	0.13 (0.05–0.33)	<0.001
Overweight and obesity (body mass index ≥23)	1.78 (1.04–3.03)	0.034	1.27 (0.64–2.50)	0.491
Serum LDL-C level at admission (mmol/L)	0.70 (0.55–0.90)	0.005	0.58 (0.42–0.81)	0.001
History of using statin before this admission	0.60 (0.35–1.01)	0.053	0.48 (0.24–0.99)	0.048
Receiving percutaneous coronary intervention	1.13 (0.67–1.90)	0.646	1.00 (0.49–2.05)	0.999
High-intensity statins prescribed at discharge (vs. non-high-intensity statin)	1.94 (1.14–3.30)	0.015	2.56 (1.23–5.35)	0.012
Frailty	0.57 (0.33–0.10)	0.050	0.75 (0.35–1.59)	0.447
Total number of comorbidities	1.03 (0.83–1.27)	0.819	1.10 (0.82–1.45)	0.540

LDL-C, low-density lipoprotein cholesterol.

## Data Availability

The study data is available from the corresponding author upon reasonable request.

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
