# Peer review of "Sex Difference in Control of Low-Density Lipoprotein Cholesterol in Older Patients after Acute Coronary Syndrome"

_geriatrics, 2022, doi:10.3390/geriatrics7040071_

Round 1

Reviewer 1 Report

Thank you for the opportunity to review your paper. 

In the introduction you take the European guidance as the target. Is this guideline promoted in Vietnam. I am afraid I don’t understand the healthcare service up I.e. if this is national guidance who makes sure patients reach targets, is it incentivised, are all pts eligible for free medications etc?

in the methods you mention inclusion criteria but don’t say ACS. I assume that was from you aims, how did you diagnose it? Was it ecg changes, trop onion etc?

I can see why having a lipid 3 months after discharge is useful but therefore aren’t you biasing the study as assume a lot of those discharged on statin with ACS without follow up may potentially be the most likely people to have stopped statins? Why do you include a 20% dropout if you are excluding those with no results at three months as surely you knew at inclusion? Also isn’t it 50% reduction AND an LDL less than 1.4?

I am afraid that I don’t think you powered the population for the regression analysis. You powered it to see the proportion making 1.4 not to examine the factors why. As a rule of thumb you need 100 people per variable for a regression analysis and you don’t have that.

in results I think you have far too many statistical comparisons and almost none are justified in such a small cohort. It is very interesting that a particular statin and ezetimibe regime is no more common in those at target. When you say LDL target though are you referring to either or and…. Ie.e. Either 50% reduction or an ldl less than 1.4….? I can actually see later it is and/or… should it be though?

do you believe the drug bit? Do you think people were lying about taking statins? You AST ranges for elevation are identical. Did you have starting AST to compare it to?

discussiojpn the gender bit is confused as you are talking about not meeting goals and mixing types of study e.g. are women undertreated? That is the message for other studies I.e. physicians still see CVD as a disease of men and under treatment women. Post menopausal Hypercholesterolaemia also mean women will have a higher baseline so may have a 50% reduction but end up with a higher LDL than men.

overall I think this is interesting but I think it is very much underpowered and there are far too many statistical comparisons. Personally I would present this as an audit of how many patients in This cohort meet LDL target at 3 months, discuss how this is already biased by people attending at 3 months. You can describe the characteristics of the cohort as a whole. You could… have a secondary aim that as we know CVd is a major killer and now killing more women than men, women, particularly Asian women, have been reported to be most at risk of side effects, post menopausal hypercholesterolaemia you would then quickly see if more women than men failed to reach target… then discuss around these points. These are a posteriori hypotheses though but you can state that plainly in the abstract.mat the moment you aim to do logistic regression seems flawed based on the number of people.

hope that makes sense but you may want to consult statistician in case you disagree.

I also see no PCSK9i use, fibrates, fish oils, bile acid resins, inclisiran etc… Again I know nothing about healthcare in Vietnam so a more comprehensive introduction would be great I.e. you use EAS targets but are not allowed to use PCSK9i etc… therefore it shouldn’t be a surprise few people meet targets.

Author Response

Thank you very much for your helpful comment and suggestions. We have tried to address your comments as much as we can. Please see our responses as follows. 

In the introduction you take the European guidance as the target. Is this guideline promoted in Vietnam? I am afraid I don’t understand the healthcare service up I.e. if this is national guidance who makes sure patients reach targets, is it incentivised, are all pts eligible for free medications etc?

Response: Thank you for your comment. We have decided to re-analyse the data using the current Guidelines for the diagnosis and management of dyslipidaemia from the Vietnam Heart Association (issued in 2015), which recommended an LDL-C goal of less than 1.8 mmol/l (<70 mg/dl). This is more contemporary to the studied period. Please see the updated results in the revised manuscript.

In the methods you mention inclusion criteria but don’t say ACS. I assume that was from you aims, how did you diagnose it? Was it ecg changes, trop onion etc?

Response: We have added the criterion of being diagnosed with ACS during this admission. Please see lines 63-64.

I can see why having a lipid 3 months after discharge is useful but therefore aren’t you biasing the study as assume a lot of those discharged on statin with ACS without follow up may potentially be the most likely people to have stopped statins? Why do you include a 20% dropout if you are excluding those with no results at three months as surely you knew at inclusion?

Response: We have removed that statement about having record of lipid profile to avoid confusion. The participants who were lost during follow up were those that did not have records of lipid profile at 3-month. Instead, we clarified this in the Result section (lines 137-138):There were 264 participants that met the selection criteria, 32 participants were lost during the 3 months of follow up. Therefore, 232 participants were included in the analysis”

Also isn’t it 50% reduction AND an LDL less than 1.4?

Response: Thank you for this comment. We have decided to re-analyse the data using the current Guidelines for the diagnosis and management of dyslipidaemia from the Vietnam Heart Association (issued in 2015), which recommended an LDL-C goal of less than 1.8 mmol/l (<70 mg/dl).

I am afraid that I don’t think you powered the population for the regression analysis. You powered it to see the proportion making 1.4 not to examine the factors why. As a rule of thumb you need 100 people per variable for a regression analysis and you don’t have that.

Response: The rule of thumb is at least 10 events (not 100) per variable. So with the number of event of 131 (131/232 of the participants achieved LDL-C target after 3 months) in our study, we can appropriately conduct regression analysis with up to 13 variables in the model (our model in Table 4 contains 11 variables). (Reference: Eric Vittinghoff, Charles E. McCulloch, Relaxing the Rule of Ten Events per Variable in Logistic and Cox Regression, American Journal of Epidemiology, Volume 165, Issue 6, 15 March 2007, Pages 710–718, https://doi.org/10.1093/aje/kwk052)

In results I think you have far too many statistical comparisons and almost none are justified in such a small cohort. It is very interesting that a particular statin and ezetimibe regime is no more common in those at target. When you say LDL target though are you referring to either or and…. Ie.e. Either 50% reduction or an ldl less than 1.4….? I can actually see later it is and/or… should it be though?

Response: Please refer to our response above. We have decided to re-analyse the data using the current Guidelines for the diagnosis and management of dyslipidaemia from the Vietnam Heart Association (issued in 2015), which recommended an LDL-C goal of less than 1.8 mmol/l (<70 mg/dl). We agree that there are too many statistical comparisons and hence, we have revised the result section. Please see the revised results.

Do you believe the drug bit? Do you think people were lying about taking statins? Your AST ranges for elevation are identical. Did you have starting AST to compare it to?

Response: We think participants were honest with their answers. We did justify in the limitation part that in this study, data were not available on medication adherence to statins (Line 244). We agree that there are too many statistical comparisons and given the incomplete data of the side effects (including liver enzymes), we have removed the section of “Side effects of statins”.

Discussion the gender bit is confused as you are talking about not meeting goals and mixing types of study e.g. are women undertreated? That is the message for other studies I.e. physicians still see CVD as a disease of men and under treatment women. Post menopausal Hypercholesterolaemia also mean women will have a higher baseline so may have a 50% reduction but end up with a higher LDL than men.

overall I think this is interesting but I think it is very much underpowered and there are far too many statistical comparisons. Personally I would present this as an audit of how many patients in This cohort meet LDL target at 3 months, discuss how this is already biased by people attending at 3 months. You can describe the characteristics of the cohort as a whole. You could… have a secondary aim that as we know CVd is a major killer and now killing more women than men, women, particularly Asian women, have been reported to be most at risk of side effects, post menopausal hypercholesterolaemia you would then quickly see if more women than men failed to reach target… then discuss around these points. These are a posteriori hypotheses though but you can state that plainly in the abstract. At the moment you aim to do logistic regression seems flawed based on the number of people. hope that makes sense but you may want to consult statistician in case you disagree.

Response: Thank you for your suggestion. We have added a secondary aim of examining sex difference in LDL-C target achievement and revise the manuscript accordingly.

I also see no PCSK9i use, fibrates, fish oils, bile acid resins, inclisiran etc… Again I know nothing about healthcare in Vietnam so a more comprehensive introduction would be great I.e. you use ESC targets but are not allowed to use PCSK9i etc… therefore it shouldn’t be a surprise few people meet targets.

Response: PCSK9 inhibitors is not available in Vietnam. Regarding the targets, we have decided to re-analyse the data using the current Guidelines for the diagnosis and management of dyslipidaemia from the Vietnam Heart Association (issued in 2015), which recommended an LDL-C goal of less than 1.8 mmol/l (<70 mg/dl).

Reviewer 2 Report

The study is focused on the success of statin treatment for LDL-C reduction in 60+ people.

The work is well written, and the results are of clinical importance. I have a few comments/questions: 

Table 1 - typo mistake - secondary lifestyle - sedentary?

The course of study is not clear - after three months, only 14 patients were on stat+ezetimibe, although, if monotherapy is not satisfactory, after 4-6 weeks, a combination with ezetimibe and/or PCSK9 inhibitor should be recommended. Please, clarify, why only 14? 

How authors explain that a higher proportion (almost 57%) in the group with "not achieved goal" had in the history use of the statins compared to almost 45 % in the group with "achieved goal". Can not it be interpreted that this group contained more persons resistant to statins therapy before the experiment (p was 0.084, so not so far from significant)? There is not discussion about this result. 

Results show that in women was a higher risk of non-successful statins therapy. It is generally well known that postmenopausal women are at higher risk of cardiovascular events. Can not be authors' results related to the postmenopausal hormonal supplementation? 

Author Response

Thank you very much for your helpful comment and suggestions. We have tried to address your comments as much as we can. Please see our responses as follows.

Table 1 - typo mistake - secondary lifestyle - sedentary?

Response: It is a typo mistake. We have made the correction (please see Table 1)

The course of study is not clear - after three months, only 14 patients were on stat+ezetimibe, although, if monotherapy is not satisfactory, after 4-6 weeks, a combination with ezetimibe and/or PCSK9 inhibitor should be recommended. Please, clarify, why only 14? 

Response: This is an observational study. The choice of lipid lowering drugs and dosage depended on the doctors that treated the participants.

How authors explain that a higher proportion (almost 57%) in the group with "not achieved goal" had in the history use of the statins compared to almost 45 % in the group with "achieved goal". Can not it be interpreted that this group contained more persons resistant to statins therapy before the experiment (p was 0.084, so not so far from significant)? There is not discussion about this result. 

Response: Thank you for your suggestion. Our new analysis with the LDL-C goal of less than 1.8 mmol/l showed that this association was significant (p=0.048 in the adjusted model, Table 4). We have added this in the discussion (lines 294-295): “More studies are needed to understand statin intolerance, statin resistance and medication adherence in this group of patients”

Results show that in women there was a higher risk of non-successful statins therapy. It is generally well known that postmenopausal women are at higher risk of cardiovascular events. Can not be authors' results related to the postmenopausal hormonal supplementation? 

Response: Thank you for the suggestion. We have added more details related to postmenopausal hormonal changes in the manuscript.

Round 2

Reviewer 1 Report

Thank you for all your changes, that is great and I learnt a lot about power calculation for regression analyses.

Re the ACS diagnosis in the methods, thank you for adding that. I still think you need to add more detail as you say inclusion criteria were those diagnosed with ACS - does ACS mean trop positive, or ECG positive, or both and those with angina but no ECG and trop changes? A definition would really help to clarify as ACS is a bit of a 'catch all' term.

I really appreciate your changes, it reads better I think. I think figure 1 is peculiarly powerful as it seems to suggest that the health disparity on presentation is worsened by treatment i.e. the amount of LDL conc overlap reduces as the men seem to be getting better care/respond better. that is really interesting.

do you think the ones on statin at admission who then failed to meet targets are familial hypercholesterolaemia patients? They may be treated for life on statins but still have lipid concentrations similar to others?

All minor stuff, think it is really thought provoking, thank you.

Author Response

Thank you for all your changes, that is great and I learnt a lot about power calculation for regression analyses.

Response: We would like to thank you for the time spent reviewing our manuscript and for your useful comments, which has helped improve our manuscript a lot.

Re the ACS diagnosis in the methods, thank you for adding that. I still think you need to add more detail as you say inclusion criteria were those diagnosed with ACS - does ACS mean trop positive, or ECG positive, or both and those with angina but no ECG and trop changes? A definition would really help to clarify as ACS is a bit of a 'catch all' term.

Response: We have added more details in the Method section (lines 77-79 and lines 105-112):

Acute coronary syndromes (ACS) refer to these 3 conditions:

(1)        ST-segment elevation myocardial infarction (STEMI): patients with acute chest pain and persistent (>20 min) ST-segment elevation on ECGs, and with troponin changes

(2)        Non-ST elevation myocardial infarction (NSTEMI): patients with acute chest discomfort but no persistent ST-segment elevation on ECGs, and with troponin changes

(3)        Unstable angina: unstable angina is defined as myocardial ischemia at rest or on minimal exertion in the absence of acute cardiomyocyte injury/necrosis (no troponin changes), and with-out persistent ST-segment elevation

Ref: Collet JP et al. ESC Scientific Document Group. 2020 ESC Guidelines for the management of acute coronary syndromes in patients presenting without persistent ST-segment elevation. Eur Heart J. 2021 Apr 7;42(14):1289-1367. doi: 10.1093/eurheartj/ehaa575.

I really appreciate your changes, it reads better I think. I think figure 1 is peculiarly powerful as it seems to suggest that the health disparity on presentation is worsened by treatment i.e. the amount of LDL conc overlap reduces as the men seem to be getting better care/respond better. that is really interesting.

Response: Thank you!

Do you think the ones on statin at admission who then failed to meet targets are familial hypercholesterolaemia patients? They may be treated for life on statins but still have lipid concentrations similar to others? All minor stuff, think it is really thought provoking, thank you.

Response: We think that familial hypercholesterolaemia may be a reason, among other reasons such as low medication adherence, low economic status and access to statins…. However, we don’t have enough data to support these conclusions. We have explained this in the limitation part: “Data were not available on medication adherence to statins. There was also limited information on other relevant factors related to LDL-C target achievement such as economic status and access to statins, the use of other medicines that can have an influence on lipid levels, and familial hypercholesterolemia”.